# Recent Efforts to Recover *Armeria berlengensis,* an Endemic Species from Berlengas Archipelago, Portugal

**DOI:** 10.3390/plants10030498

**Published:** 2021-03-07

**Authors:** Teresa Mouga, Susana Mendes, Inês Franco, Ana Isabel Fagundes, Nuno Oliveira, Paulo Crisóstomo, Lurdes Morais, Clélia Afonso

**Affiliations:** 1MARE—Marine and Environmental Sciences Centre, ESTM, Polytechnic of Leiria, Edifício CETEMARES, Av. Porto de Pesca, 2520-641 Peniche, Portugal; susana.mendes@ipleiria.pt (S.M.); clelia@ipleiria.pt (C.A.); 2ESTM, Polytechnic of Leiria, 2520-641 Peniche, Portugal; inesmafranco@hotmail.com; 3SPEA—Sociedade Portuguesa para o Estudo das Aves, Avenida Columbano Bordalo Pinheiro, 87 – 3° Andar, 1070-062 Lisboa, Portugal; isabel.fagundes@spea.pt (A.I.F.); nuno.oliveira@spea.pt (N.O.); 4ICNF—Instituto da Conservação da Natureza e das Florestas, Reserva Natural das Berlengas. Av. Mariano Calado, 57, 2520-224 Peniche, Portugal; paulo.crisostomo@icnf.pt (P.C.); lurdes.morais@icnf.pt (L.M.)

**Keywords:** conservation, endangered species, flora, island, rupicolous species, threats

## Abstract

Berlengas archipelago is a UNESCO world heritage site and the only location where *Armeria berlengensis* is found. This species faces various threats, namely, human disturbance, the presence of *Carpobrotus edulis,* yellow-legged gull, common-rabbit, and black-rat populations. Thus, exclusion areas were installed, which blocked the access of most Gulls, aiming to promote the recovery of *A. berlengensis*. Additionally, rabbits and rats were removed from the island. After six years of surveys, there has been an increase in the number of individuals of *A. berlengensis* in the exclusion areas, and a clear shift in the size structure of the *A. berlengensis* population. Significant changes in the height and diameter of the individuals were also noted. These findings indicate that the population of *A. berlengensis* is changing and becoming a healthier population. Principal component analysis results show a straightforward dissimilarity between the areas with *A. berlengensis* and those without the species and allowed the clustering of two groups: the rupicolous species and the nitrophilous species. *A. berlengensis* produces few seeds (seed set 3.4%), which raises concern regarding the long-term survival of the species. Thus, further conservation efforts must be implemented, such as the control of invasive species, gulls, and ruderals, to allow for the recovery of *A. berlengensis*.

## 1. Introduction

The Berlengas archipelago was firstly classified as a natural reserve in 1981. In 1998, this area was classified as a marine reserve area, and its territory increased to its present size (terrestrial surface 725.6 ha, marine area of 17,776.7 ha). In 2011, the Berlengas became part of the list of UNESCO world heritage sites, which aim to preserve representative terrestrial and marine ecosystems of the Portuguese coast. The Berlengas archipelago hosts several important species and has other interesting features, which are relevant in the national and international context. These unique features of the archipelago are due to its insular nature, geological characteristics, geographical location, and climate, along with historic low levels of human interference. All together, these characteristics have contributed to the preservation and speciation of the terrestrial and marine flora and fauna [1,2].

Oceanic islands are known for their high levels of plant diversity, due to disjunct geographical distribution that leads to speciation [3,4]. The main factors contributing to genetic speciation include the creation of a barrier within a previously widley distributed taxon [5] and the limited dispersal of seeds, which favors genetic differenciation and, thus, fosters rapid speciation [6]. In the Berlengas archipelago, this speciation was due to island isolation from the continent that occurred after the Wurm Glatiation, ca. 18,000 B.C. [7]. Owing to the insular isolation, and to very harsh climatic and edaphic conditions, most of the botanical species are rupicolous aero-halophytic plants that can tolerate the presence of salt and action of strong winds, simultaneously. The terrestrial flora evolved differently from the mainland species, as these plants struggle to survive in extreme summer dryness, very thin soils, salty environments, and strong regular north-westerly and intense south-westerly winds. Most of the plants are thicker and smaller in size than their continental counterparts, with smaller leaves, which prevent wind damage and excess evaporation. These harsh conditions led to speciation processes that differentiated three endemic species of great conservation value: *Armeria berlengensis* Daveau, *Herniaria berlengiana* (Chaudhri) Franco and *Pulicaria microcephala* Lange [8,9,10]. As most island endemic plants, these species have a small number of individuals and a low distribution, thus, they are particularly susceptible to extinction [4].

Plant survival and population fitness vary according to environmental factors and to human interference [11]. Endemic island plant species from the Mediterranean area are generally threatened by urbanization, touristic activities, fire, invasive alien species, changes in agriculture practices and collecting pressure [4]. The endemic and halophytic flora of the Berlengas Archipelago is also threatened by these factors, as observed in many other European coastal territories which host threatened endemics with very limited areas of distribution [4,11,12,13,14]. First, increasing tourism activities during recent decades have led to the disturbance of the ground in recreational areas. Additionally, the introduction—in the late 1950s—of the African species *Carpobrotus edulis* (L.) N.E.Br., in an attempt to reduce rockfalls in recreational areas has also led to declines in native species. This invasive species spread out over the cliffs and hillsides [15], reaching a total area of 38,533 m^2^ in Berlenga Grande, which represented around 37% of its terrestrial area [16]. Other species, such as the ruderals *Calendula suffruticosa* subsp. *algarbiensis* (Boiss.) Nyman, *Cerastium glomeratum* Thuill., *Echium rosulatum* Lange, *Fumaria muralis* Sond. ex Koch, and *Hyoscyamus albus* L., *Linaria amethystea* subsp. *multipunctata* (Brot.) Chater & D.A. Webb, *Nicotiana glauca* Graham, tend to flourish in more disturbed areas, namely, where human intrusion is stronger, such as along paths and construction sites [17]. Furthermore, the increase in the amount of phosphorus and nitrogen produced by the large population of yellow-legged gull (*Larus michahellis)* that breeds on the Berlengas Archipelago has caused significant soil changes, namely, increases in water and nutrient retention, which increases organic matter, and decreases the soil pH [18]. These soil changes have considerably altered the natural floristic composition, promoting the emergence of annual nitrophilous vegetation and the decline of autochthonous perennial species [17,19,20,21,22]. Finally, introduced herbivores—the black-rat (*Rattus rattus*) and common-rabbit (*Oryctolagus cuniculus*)—have also played a role in decreasing the natural vegetation. It is well known that extensive grazing increases open land habitats, thus, improving grasslands and causing the decline of shrublands [23]. Grazing on perennial plants by rabbits, which also profusely excavate the thin soil, has put extensive pressure on *A. berlengensis* [1]. As for rats, these are omnivores that feed on vertebrates, invertebrates, fungi and plants, but not on *A. berlengensis*, as stated by Nascimento, T. [24]. Yet, *Carpobrotus* spp. fruits have been found to be eaten and dispersed by both rabbits and rats, thus contributing to *C. edulis* success [25]. Additionally, gull-derived resources are significant for both rodent species, and these have altered their growth rates, reproductive output, and population densities [26].

In short, the species *A. berlengensis* has changed in status from vulnerable [15] to critically endangered, according to the International Union for Conservation of Nature (IUCN)threat classification scheme, due to the presence of *C. edulis*, herbivores, and *L. michahellis* that nest over the most mature plants [27,28].

*A. berlengengis* is registered in Annex II of the Habitats Directive, due to their conservation relevance [28,29]. Therefore, to preserve important habitats and species, the control of *L. michahellis* has been undertaken for more than twenty years—through egg removal—to control the size of the nesting population on the island. Additionally, between 2016 and 2018, black-rat eradication campaigns were implemented and in 2017 and 2018, rabbits were also eliminated from the Berlengas main island. In addition, in 2014, conservation actions commenced that permitted the removal of more than 90% of *C. edulis* from the main island [16]. Finally, in May 2015, several areas with metal structures and fishing lines were installed (exclusion areas), which blocked the access of most gulls. The vegetation of these areas has been seasonally monitored between 2015 and 2020 to assess vegetation evolution. Spring data were analyzed during the study period to evaluate vegetation variation during the most productive season on the island. Seed set was also measured to better understand the natural reproduction rates of *A. berlengensis*.

This study, therefore, aimed to determine the effectiveness of various conservation strategies on the population of *A. berlengensis*. As such, we assessed the effectiveness of the establishment of exclusion zones on the growth promotion of *A. berlengensis,* on the growth of the other rupicolous species, and on the growth of nitrophilous species. The height and diameter of *A. berlengensis* was also determined annually to evaluate the growth of the individuals and the size structure of the *A. berlengensis* population in the studied areas. Moreover, we evaluated the number of seeds produced by *A. berlengensis*—by seed cluster—to estimate the seed set of the species, in attempt to understand the recovery capacity of the species.

## 2. Results

### 2.1. Armeria Berlengensis Surveys

After six years of surveying the exclusion and control areas, there is clear evidence of the nonexistence of *A. berlengensis* organisms in all the areas where it was initially absent (all B areas), except for one dead adult individual observed in 2015 (height 30–40 cm). This individual was excluded from the following surveys.

Yet, in the areas were *A. berlengensis* initially existed, a significant difference was noticeable (t-student test, *p*-value = 0.013) between the number of individuals growing in the exclusion areas and the control areas (Figure 1), demonstrating the heterogeneity of the data.

When only analyzing the total number of individuals of *A. berlengensis* in the different areas, during the Spring (2015 to 2020), when the species grows and blossoms, we can see that this value ranged from 85 to 201 in the exclusion areas A(1–3) and from five to 17 for the control areas A(4–6). Thus, a well-defined trend of growth in the number of individuals can be seen, mainly in 2020 (Table 1).

When assessing the differences between the sampling periods, for each type of structure (namely, exclusion areas A(1–3) “structure + *Armeria*” and the control areas A(4–6) “no structure + *Armeria*”), it was observed that these are not statistically significant (ANOVA_A(1–3)_, *p*-value >0.05 and Kruskal–Wallis_A(1_–_6)_, *p*-value > 0.05).

Additionally, when comparing the type of structure (namely, exclusion areas A(1–3) “structure + *Armeria*” and the control areas A(4–6) “no structure + *Armeria*”), for each sampling period there were no statistically significant differences for the number of individuals of *A. berlengensis* (t-student test, *p*-value > 0.05 for all cases). When analyzing the data globally, that is, when comparing the number of individuals of *A. berlengensis* for the six sampling periods, the results also show the same pattern, that is, no statistically significant differences (ANOVA, *p*-value > 0.05) (Figure 2).

Between 2015 and 2020 there was a clear increase in the number of plantlets, mainly in 2019 for the exclusion area A(1–3) (Figure 3a) and in 2020 for both exclusion areas A(1–3) and control areas A(1–6) (Figure 3b). It was possible to observe an increasing trend in the number of plantlets, mostly after 2019, when the total number of plantlets surpassed the number of juvenile plants, for the first time, during our study. The other categories remained rather stable, showing minor fluctuations between the years. Moreover, the very low number of large plants (>41 cm) in all the areas surveyed was very noticeable. However, these plants are of large dimensions and take up considerable space in the study area.

Regarding the mean height and mean diameter of the plants, as expected, we can see a clear correlation between both measures, for all the years and both exclusion area A(1–3) and control A(1–6) (R^2^_A(1–3)_ “structure + *Armeria*” = 0.688, *p*-value < 0.01 and R^2^_A(4–6)_ “no structure + *Armeria*” = 0.760, *p*-value < 0.01; Figure 4), indicating that the increase in height is accompanied by an increase in width.

As for the height of *A. berlengensis*, it varied between 11.05 cm in 2020, corresponding mainly to plantlets and small adults, for the exclusion areas A(1–3) and 25.83 cm in 2019, for the control areas A(4–6), corresponding to larger adult plants (Figure 5a). The mean values of the plants in the exclusion area A(1–3) was 15.54 cm and in the control areas was 19.39 cm. The diameter of *A. berlengensis* varied from 18.7 cm in 2020, corresponding mainly to plantlets and small adults, for the exclusion areas A(1–3) to 42.50 cm in 2017, for the control areas (A4–6), also corresponding to larger adult plants. The mean diameter of the plants in the exclusion areas A(1–3) was 26.19 cm and in the control areas A(1–6) was 32.25 cm (Figure 5b).

Additionally, when comparing the height between the six sampling periods (separately by type of area, namely, for the exclusion areas A(1–3) “Structure + *Armeria*” and the control areas A(1–6) “No structure + *Armeria*”), the results did not show statistically significant differences, for the control areas A(1–6) “No structure + *Armeria*” (ANOVA, *p*-value > 0.05). However, when the comparison was made for the exclusion areas A(1–3) “Structure + *Armeria*”, the results showed statistically significant differences when comparing the sampling period (Kruskal–Wallis, *p*-value < 0.001; Figure 5a). More specifically, differences can be observed when comparing the spring of 2020 with all the others (Dunn, *p*-value < 0.001), as well as the Spring of 2019 with that of 2017 (Dunn, *p*-value = 0.045) and 2015 (Dunn, *p*-value = 0.003) (Figure 5a).

For the diameter, the results follow a similar pattern, as they showed statistically significant differences when comparing the sampling periods, only for the exclusion areas (Kruskal–Wallis_A(1–3)_ “Structure + *Armeria*”, *p*-value < 0.001; Figure 5b) and Kruskal–Wallis_A(1–6)_ “No structure + *Armeria*”, *p*-value > 0.001). In particular, for the exclusion areas, the observed differences were between the Spring 2020 and all others (Dunn, *p*-value ≤ 0.001;), as well as between Spring 2019 and Spring 2016 to 2018 (Dunn, *p*-value < 0.05) and, finally, between Spring 2015 and also Spring 2016 to 2018 (Dunn, *p*-value < 0.05) (Figure 5b).

There is a decreasing trend in height and diameter that occurred mainly after 2018 in the exclusion areas A(1–3) (Figure 6,c), but this trend did not occur in the control areas A(1–6), where these values are much more heterogeneous (Figure 6b,d). This is concomitant with the increase in the number of individuals previously discussed (Figure 3a).

It is also worth mentioning that there are many outliers in the exclusion areas A(1–3) all corresponding to large adult plants, presenting greater height and/or larger diameter than the median values. These represent the oldest individuals of the surveys, also mentioned above in Figure 3.

### 2.2. PCA Analysis

The principal component analysis (PCA) was applied to the species coverage on different sampling periods. The PCA results showed that 51.6% of the total variance was explained by the two principal components. Additionally, PC1 explained 36.7% of the variance of the original variables analyzed, which is the most significant response regarding the clear separation between the areas with or without *A. berlengensis* (Table 2 and Figure 7).

Having clearly defined two clusters, the PCA results demonstrated that there is a straightforward dissimilarity between the areas with *A. berlengensis* and the areas without *A. berlengensis* (Figure 7). The areas A (A1 to A6) presented *A. berlengensis* during the entire period of the study while the areas B (B1 to B6) failed to regain any individuals of *A. berlengensis*, either in the exclusion or the control areas.

Regarding the species coverage (%) pattern, in the CP1 positive axis, there is a cluster that includes *A. berlengensis* (sp1), the most characteristic rupicolous species on the island, alongside *Plantago coronopus* L. (sp7), and *Spergularia rupicola* Lebel ex Le Jol. (sp9) (Figure 8). Additionally related to *A. berlengensis*, are the different species from the Poaceae family (sp4) and *Polycarpon alsinifolium* (Biv.) DC. (sp14).

In the CP1 negative axis, there is another cluster which includes the ruderal nitrophilous species, such as *C. suffruticosa* subsp. *algarbiensis* (sp3) and *Echium rosulatum* (sp5). Additionally, the pattern observed showed that these two species are strongly correlated, which means that the increasing abundance of *C. suffruticosa* subsp. *algarbiensis* is accompanied by the increasing abundance of *E. rosulatum*, and that the environmental conditions that benefit one also benefit the other. *Lobularia maritima* (L.) Desv. subsp. *maritima* (sp6) seems to be correlated with the former two species, but to a smaller extent, also being a ruderal species. Similarly, but less markedly, species such as *Mercurialis ambigua* L. (sp13), *Erodium cicutarium* (Cav.) Tourlet (sp10), *Scrophularia sublyrata* Brot. (sp11), and *Urtica membranacea* Poir. (sp2), likewise showed an association with each other, all being nitrophilous species. As for *Atriplex prostrata* Boucher ex DC. in Lam.t DC. (sp8), again a nitrophilous species, this species is allocated along the second axis, meaning that its growth is not affected by the distribution of rupicolous vegetation nor by the distribution of ruderals vegetation.

The remaining species showed a low relevance behavior since their variability is reduced, as vectors are closer to the origin (Figure 8), showing a slight contributory power for the global behavior of the species under analysis.

### 2.3. Armeria berlengensis Seed Set

During our survey, 98 fruit clusters were analyzed, with a mean number of fruits per cluster of 48.17. The results showed that the total number of seeds produced by *A. berlengensis* is very low, having been collected only 164 seeds out of 4721 calyxes. There is a mean value of 1.67 seeds per fruit cluster and a seed set of only 3.47% (Table 3).

More specifically, by observing 19 infructescences, it should be noted that the low seed set observed (3.39%) was mainly due to the presence of non-fertilized ovaries, which accounted for more than 50% of the total number of flowers analyzed (Table 4). Additionally, many calyxes were empty (30.53%), and more than 15% were found that were already decaying.

## 3. Discussion

The results demonstrated that there is a rising number of individuals growing in all the studied areas A(1–6), but not where the species *A. berlengensis* was previously absent, such as in areas B, the exclusion areas B(1–3) or the control areas B(4–6). This means that the capacity of recruitment of the species is rather limited and that the presence of the structures installed in A(1–3) and B(1–3) is efficient in increasing the number of individuals in the vicinity of sites where there are individuals of *A. berlengensis*, but not beyond these A(1–3) areas. This can be explained by the fact that *Armeria* seeds are wind dispersed and the distance of seed transportation is low, as shown by Philipp et al. [30] for *A. maritima* (maximum distance of 3.5 m, mean distance 0.88 m). Therefore, during the six years of the surveys, we were unable to expand the distribution area of *A. berlengensis*.

Additionally, it was clear that from the first years of the study and until 2018, the number of new individuals was very low, with a reduced number of individuals under 10 cm, when compared to the older individuals (especially young adults). Thus, for the first few years of the survey, the pyramid was truncated at its base, meaning that the reproduction and recruitment of the species seemed to be compromised [31], and so the survival status of the species seemed to be at risk. However, this pattern changed in the surveys from 2019 and 2020, when the recruitment of new plantlets augmented substantially, mainly in the exclusion areas A(1–3). Thus, the pyramid base also grew considerably, evolving into a much more equilibrated and healthy population, with a higher number of individuals belonging to younger classes, due to recruitment of new plants by germination [32]. The increase in the number of *A. berlengensis* plantlets in the control area is also noticeable, but to a much less extent than in the exclusion areas.

Due to the windy habitat it is exposed to, *A. berlengensis* presents a prostrate, pulviniform shape; therefore, the individuals are usually wider than they are tall, in line with most of the littoral shrubs [33]. Thus, the correlations found between diameter and height are to be expected, because as they grow, the plants become successively wider and taller.

The significant differences found for the diameter and the height of the individuals of *A. berlengensis* in the exclusion areas A(1–3), can be explained by the recruitment of new individuals, occurring mainly in 2019 and 2020, as already discussed. New individuals are smaller in size and diameter, and thus, are responsible for the decreasing trend in diameter and height that occurred mainly after 2018. The significant differences observed both for diameter and height between 2020 and the other years is meaningful, for it is coincident with the increase in the number of new plantlets observed growing in the studied areas.

The removal of herbivorous mammals from the island had a beneficial effect on the native vegetation recovery, as has been widely documented by many authors worldwide [34,35,36,37,38,39,40,41]. We believe that the growth of new plantlets is due to the removal of the mammals.

The natural vegetation of Berlengas is chasmophytic, mainly composed of geophytes, hemicryptophytes, and chamaephytes, often covered by marine salt spray [42]. Therefore, the species *A. berlengensis, N. bulbocodium, S. rupicola, S. sublyrata*, and *A. pachycharpa*—all rupicolous species—should thrive on the island, mainly in undisturbed niches. Additionally, the association between *A. berlengensis* and *S. rupicola* has been acknowledged before, as they are specialized species that grow in granitic rock crevices and on cliffs, and they are well adapted to harsh environmental conditions [33]. The PCA results reinforce the idea that the presence of *A. berlengensis* is accompanied by other species that are adapted to rocky environments and scarce soil. Such species include: *Plantago coronopus*, which is a characteristic species of dry grasslands with scarce soil; *Spergularia rupicola*, a species that grows in rocky places, near the shore [9]; *Polycarpon alsinifolium*, a species that grows in sandy habitats [43], and species from the Poaceae family. All these species are adapted to a thin soil layer and low nutrient content. Additionally, the exclusion area played no part in the recovery of the *A. berlengensis* population in areas where the species was already absent, as already stated, with the principal component PC1 clearly defining two very distinct areas. This means that the increase in the *A. berlengensis* population relies on the species current distribution area and may depend on the diminishing impact of the birds on the plateau. As stated, this may be due to the low distance seed dispersal.

The differences observed between the exclusion and the control areas are due to a decrease in the inflow of gull droppings in the exclusion zones. These are known to be rich in nitrate and phosphate nutrients [29,44,45], which causes significant ruderalization of the vegetation [20,46]. Thus, the presence of some abundant ruderal species in the exclusion zones indicates that there is also disturbance in these areas, which reflects the presence of N-enriched soil, and/or the presence of gulls [21] before the installation of the structures. There is evidence that this part of the island (plateau) has been altered considerably due to the presence of gulls which have transported new annual and ruderal plant species to the island that thrive on this altered N-enriched soil [47]. Gulls must, therefore, be the main bottleneck to the growth of rupicolous vegetation on the island. Therefore, the presence of the structures on the exclusion zones, which discourage the occurrence of gulls, seems to be playing a significant role in the recovery of this type of vegetation. It is expected that in the coming years, in the exclusion zones, an increase in rupicolous vegetation will be observed as the N and P soil concentrations decrease. However, this will be a slow process.

The PCA results also confirm that *A. berlengensis* and the other rupicolous species are on the CP1 positive axis, while *C. suffruticosa* subsp. *algarbiensis* and *E. rosulatum* are on the negative PC1 axis. These are both endemic species from the littoral area of the Iberian Peninsula, more robust, and well adapted to areas of greater soil accumulation, due to the gulls’ excrements, human activity and rock weathering [17,29]. The growth of rupicolous species occurs under the opposite environmental conditions to the growth of ruderal species. The ruderals, *E. rosulatum* and *C. suffruticosa* subsp. *algarbiensis*, presented higher cover percentages in places where *A. berlengensis* was absent (B areas, either exclusion or control zones), since they rapidly occupy the available N-enriched soil. Most of the species associated with these ruderals (e.g., *L. maritima*, *M. ambigua, O. calendulae*) also handle human interference well, and thus, prosper in disturbed soil [19]. Almost 40% of the biotopes of the Berlengas archipelago are from deep soil vegetation and only 16% belong to species that are adapted to crevices and thin soil layers. Additionally, it is clear that *C. suffruticosa* subsp. *algarbiensis* is an ornithocoprophil species and its presence on the island was only mentioned in 1989 [19]. Therefore, its abundance in the Berlengas seems to be a direct consequence of the presence of the gull populations, the size of which increased dramatically since the beginning of the twentieth century until 1997, when the colony control measures were implemented [16,48]. Thus, the higher presence of these ruderal species in more disturbed soil, evidenced by PCA results, is not surprising. This is an indicator that the presence of gulls (either nitrification or soil and plant abrasion) enhances the growth of ruderal species, which has been confirmed by other authors.

Therefore, we believe that the exclusion zones, along with the absence of mammals, are allowing a slow recovery of the natural rupicolous vegetation on the plateau, where our studies were performed, but the soil is still under the influence of the gull populations, that is, it has been N-enriched, which is leveraging the growth of the nitrophilous vegetation.

The seed set found for *A. berlengensis* was very low when compared to other similar species, such as *A. pseudoarmeria* (35.2%), *A. welwitschii* (51.4%) [49] and *A. maritima* (29.3 seeds per fruit cluster) [50]. These low seed production values for *A. berlengensis*, along with the very high number of unfertilized ovaries (over 50%), indicate that pollen dispersal by insects and birds may be poor, and fertilization is probably scarce. These findings raise a concern about the survival of the species and, along with the known threats already stated, the low seed set may have been a major issue in the previous decreases in *A. berlengensis* populations and the recruitment of new plants. Despite the low number of seeds produced, the dispersal and germination of the few seeds produced by *A. berlengensis* seems to be efficient enough at short distances, as the recruitment data for 2019 and mainly for 2020 indicate that new plantlets germinate when disturbance decreases.

Further studies will confirm if the presence of the structures will allow the plantlets to grow into adult plants and if these individuals will expand further outside the exclusion areas installed. Due to the slow growth rate of *A. berlengensis,* these future results will take a fair amount of time to acknowledge extensive recovery of the population.

## 4. Materials and Methods

### 4.1. Study Area

The Berlengas Archipelago (39°24′49″ N–9°30′29″ W) is located in the Atlantic Ocean, on the Portuguese continental shelf, on the western side of the Iberian Peninsula, close to Cape Carvoeiro (Peniche). It distances approximately 10 km from the mainland and is composed of three island groups: Berlenga Grande Island and adjacent islets and reefs, Estelas Islands and Farilhões Islands (Figure 9) [1].

In May 2015, the study area was established in the plateau, where the degradation of granite rock has allowed the accumulation of a thin layer of soil. This is the preferred soil for the growth of *A. berlengensis*, and indeed, it was formerly known as the “valley of the *Armeria*”, where several herbaceous species grow, as well as the two varieties of *A. berlengensis*, as the name of the valley indicates [51]. Yet, since a survey of 2004 [29], the number of *A. berlengensis* was shown to be declining, mainly due to the presence of gull nests and the presence of herbivores (Figure 10).

The area was divided into 4 subareas, each one composed of three squares of 10 × 10 m. In two of these subareas, which we called “exclusion areas”, one presenting *A. berlengensis* individuals (A1, A2 and A3) and another without any individual of *A. berlengensis* (B1, B2 and B3) we installed structuresmade of steel poles and a fishing line, set to prevent the approach of gulls (Figure 11a). Each pole was placed along the sides of each square, evenly spaced by 1 m. The fishing lines were positioned parallel to each other, pole to pole, at about 1 m high. In total, 190 poles were used in the exclusion areas. The other subareas, again one exhibiting *A. berlengensis* growing (A4, A5 and A6) and another without any individuals of *A. berlengensis* (B4, B5 and B6) were used as controls, and thus, no structures were placed, allowing free access by the gulls. Despite the setting of the structures, one gull managed to set its nest over one of the largest individuals of *A. berlengensis*, inside one of the exclusion areas (Figure 11b). Yet, this was a unique event, which has not been recorded on any other occasion, in any other survey.

Between 2015 and 2020, species presence and cover were assessed in the Spring (15 May, 16 May, 17 May, 18 May, 19 May, 20 Jun) when most species grow and blossom. Each species cover percentage was registered in a 2 × 2 m square defined in the center of each 10 × 10 m square.

During this period, all individuals of *A. berlengensis* present in the subareas were counted and measured, both in diameter and height, as well as the size structure of the individuals in the exclusion areas. For this purpose, we divided the height of the vegetative plant into 5 categories. Plants smaller than ten centimeters were considered to be plantlets, between 11 and 20 cm young adults, and the other 3 categories were registered as adults (20–30, 30–40 and >41 cm).

Additionally, between January and May 2017, 98 fruit clusters (infructescences) from *A. berlengensis* were harvested. These clusters were collected randomly throughout the main island, from adult plants (height >30 cm) with more than 20 clusters each, 3 clusters being collected from each plant, to prevent significant loss of reproductive units. The seed set was determined based on the total number of seeds per the total number of fruit clusters harvested. Calyxes were also analyzed to determine if ovaries were present and fertilized.

### 4.2. Armeria berlengensis Morphology and Distribution

*A. berlengensis* is a sturdy bushy plant, up to 80 cm (130 cm) in diameter, and with aerial branches of variable lengths (up to 50 cm). Leaves 30–70 × 3–14 mm are arranged in terminal tufts, from linear-lanceolate to lanceolate, usually with 3 to 5 nerves, flat and fairly rigid. External bracts are oval-shaped, cusped, longer than those of the middle part, the internal ones are obovate-oblong, mucronate. The calyx is 6–9 mm, with rows of nurtured hairs, with these being more than 0.3 mm long. Corolla is pink to white [9]. Two varieties of the species have been acknowledged on the archipelago, var. *berlengensis* which has glabrous leaves and var. *villosa* with villous leaves [51,52,53]. Pollen is transported by nonspecialized insects and, less frequently, by birds [30,54]. Like all the other *Armeria* species, *A. berlengensis* seeds are adapted for wind dispersal and are dispersed enclosed in the calyx, as suggested by the parachute-like shape of the calyx [50].

The more extreme habitats are represented by the refuge of native stress-tolerant vegetation, such as *A. berlengensis* [9,16,29,42]. *A. berlengensis* occurs throughout the archipelago, mainly on rocky outcrops, halophytic cliffs and consolidated gravel pits under the influence of wave spray and strong winds (Figure 12).

### 4.3. Statistical Analysis

To study the number of *A. berlengensis* individuals in the exclusion areas A(1–3) and the control areas A(4–6), the parametric Student’s t-test was performed [55], globally, as well as within each sampling period (Spring 2015 to Spring 2020). Additionally, to compare the cover percentage across the study period (2015–2020) of *A. berlengensis* in the Spring, an analysis of variance (ANOVA) with one factor was performed. The procedure was performed globally, as well as for each type of structure (namely, exclusion areas A(1–3) “structure + *Armeria*” and the control areas A(4–6) “No structure + *Armeria*”). All data were checked for normality and homoscedasticity [55]. Whenever the assumptions were not met, the Kruskal–Wallis non-parametric test was performed. Whenever applicable, Tukey or Dunn multiple comparison tests were used (according to the ANOVA or the Kruskal–Wallis test, respectively). Additionally, to evaluate the strength of the correlation between the mean height and the mean diameter, across the study period (2015–2020) of *A. berlengensis* in the Spring, Pearson’s correlation coefficient (r) was used [55]. Results were considered significant at *p*-value ≤ 0.05 level. Where applicable, results are presented as mean ± standard deviation (SD) or mean ± 95% confidence interval (CI). All calculations were made with IBM SPSS Statistics version 27 software. Finally, principal components analysis (PCA), based on correlation matrices, was applied to identify the main associations among the cover of the observed species and the areas (with *A. berlengensis* exclusion area and control, without *A. berlengensis* exclusion area and control) and sampling periods (Spring 2015 to Spring 2020). The principal component analysis (PC1 and PC2) provides information on the most meaningful parameters, which describe a whole data set, affording data reduction with minimum loss of original information. Although the results concerning the first two components were presented, the others were also analyzed. To perform the analysis, only those species with an annual percentage of coverage higher than 2% were considered, with the remaining species being considered as a record without significant expression. Values expressed as a relative percentage were arc-sine square-root transformed before analysis [56]. All calculations were performed with the CANOCO version 4.5 package [57].

## 5. Conclusions

The survival of the natural rupicolous vegetation of Berlenga Island faces several challenges due to the disturbance the island has been subjected to in recent decades. PCA showed two distinct clusters of species, namely, natural rupicolous species and nitrophilous vegetation, showing the major influence that herbivores and gulls have had on the island’s vegetation. The installation of protective structures (exclusion zones) that prevent the access of gulls, along with the removal of rats and rabbits, seem to have played an important role on the recovery of the natural rupicolous vegetation, namely on *A. berlengensis*. The number of seedlings increased and significant differences in height and diameter were noticeable, mainly for 2020. However, in addition to its slow growth, the particularly low seed set of *A. berlengensis* hinders the rapid recovery of the species. Thus, a longer experimental period is necessary to corroborate if seedling establishment is successful and, thus, if these conservation measures are truly effective.

Although this is a preliminary study, some considerations can be already made regarding the population of *A. berlengensis*. The presence of man, gulls, and *C. edulis* in other locations of the main island, is enriching the soil nitrate and phosphate contents, which favors the growth of ruderal vegetation at the expense of rupicolous vegetation [58]. Therefore, the control of these problematic issues seems to be the best way to allow the recovery of native vegetation. On the one hand, controlling the touristic activities, human construction and trampling on the island, is of utmost importance. These measures have already been implemented in the management plan of the Berlenga Reserve, in particular, with the implementation of a carrying capacity on the island. On the other hand, continued control of the gull population, and the installation of other exclusion areas will reduce soil nitrification and gull trampling, although much of the plateau’s soil has already been degraded, hindering the growth of rupicolous species.

Finally, the complete removal of mammals will favor the growth of shrubs instead of annual vegetation, and the control of *C. edulis* will allow a recovery, even if slow, of native vegetation. Further studies of the exclusion areas and outside, of the rupicolous and the nitrophilous vegetation on the next years will provide us with more clues into the main factors influencing the already visible trend of the Berlenga vegetation.

## Figures and Tables

**Figure 1 plants-10-00498-f001:**
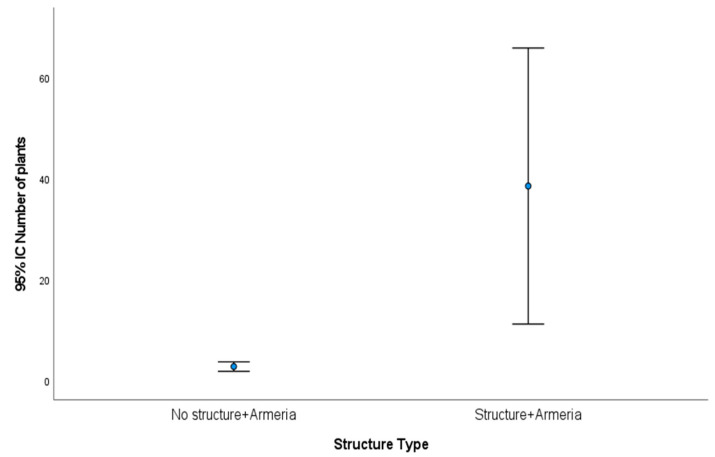
Mean number of individuals of *A. berlengensis* in the exclusion areas A(1–3) “structure + *Armeria*” and the control areas A(4–6) “no structure + *Armeria*”. Results are presented as mean ± 95% Confidence Interval (CI).

**Figure 2 plants-10-00498-f002:**
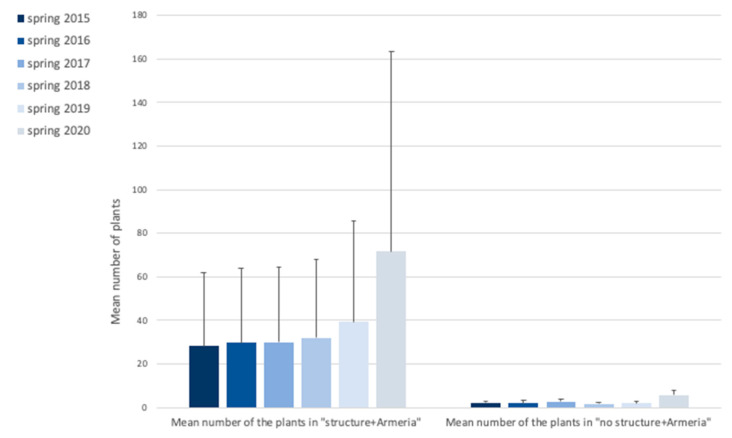
Mean number of individuals of *A. berlengensis* growing in all the areas analyzed, counted between the spring of 2015 and the spring of 2020. Results are presented as mean values ± SD.

**Figure 3 plants-10-00498-f003:**
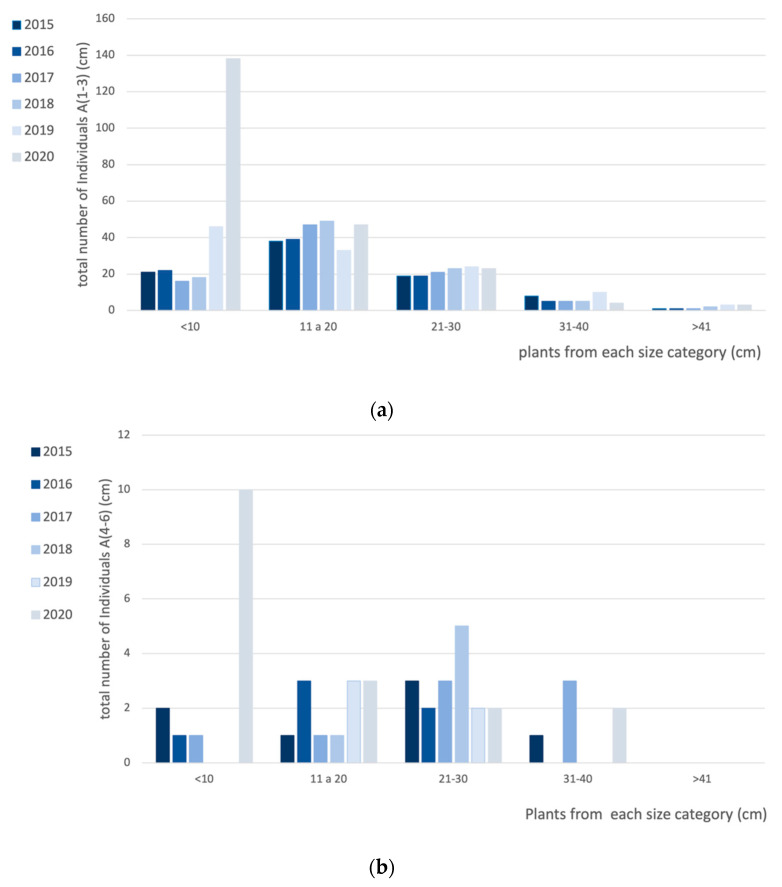
Variation of the size structure of *A. berlengensis* population in the studied areas between the Spring of 2015 and the Spring of 2020: (**a**) in the exclusion areas A(1–3) “Structure + *Armeria*” (**b**) in the control areas A(4–6),”No structure + *Armeria*”.

**Figure 4 plants-10-00498-f004:**
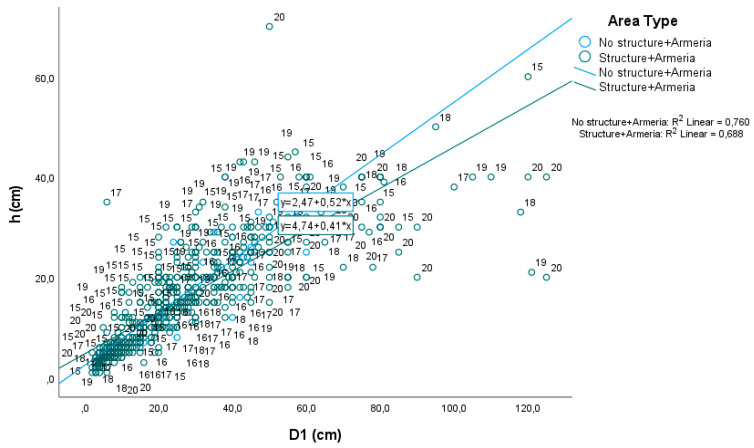
Pearson’s correlation between height (h) and diameter (D) of the individuals of *A. berlengensis*, for all the individuals measured in the exclusion areas A(1–3) “Structure + *Armeria*” and the control areas A(1–6) “No structure + *Armeria*”, between the spring of 2015 (“15”) and the spring of 2020 (“20”).

**Figure 5 plants-10-00498-f005:**
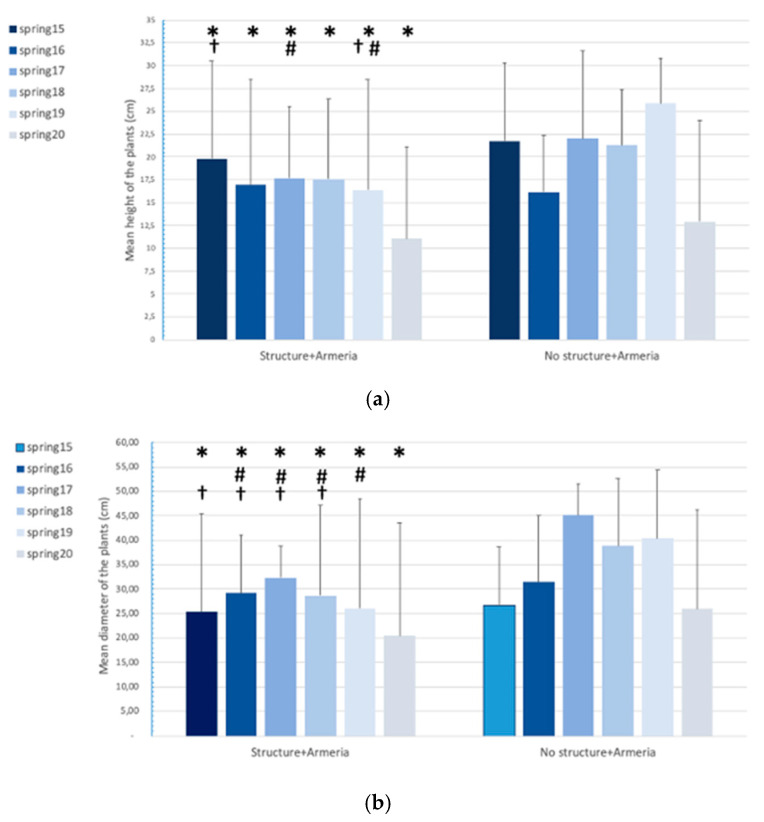
Mean **(a)** height of the plants of *A. berlengensis* present in the exclusion areas A(1–3) “Structure + *Armeria*” and in the control areas A(1–6) “No Structure + *Armeria*” and **(b)** diameter of the plants of *A. berlengensis* present in the exclusion areas A(1–3) “Structure + *Armeria*” and in the control areas A(1–6) “No Structure + *Armeria*”, between Spring 2015 and Spring 2020, in centimeters. Results are presented as mean values ±SD. Symbols above bars represent significant statistical differences between medium values (* between 2020 and all the other years; # between 2019 and other years; † between 2015 and other years).

**Figure 6 plants-10-00498-f006:**
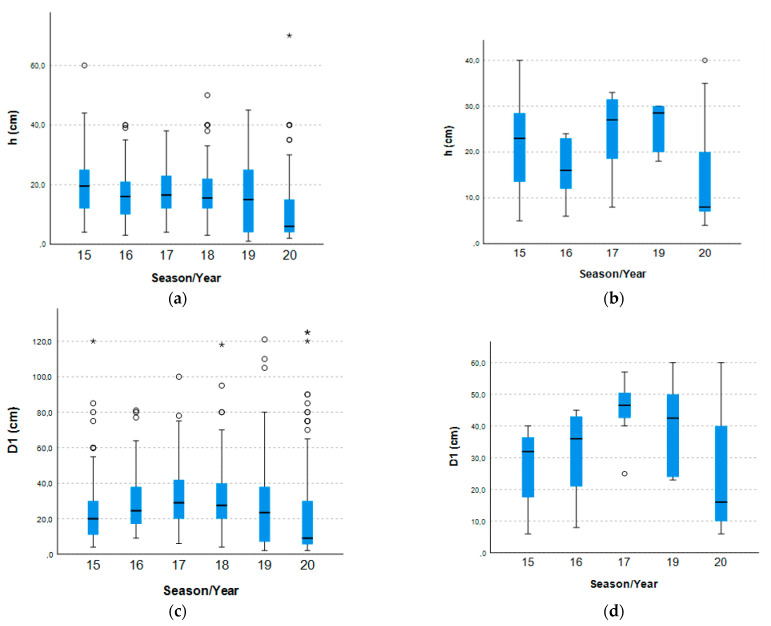
Boxplots for the median values of the height (h) and the diameter (D1), in centimeters, of the vegetation measured between Spring 2015 and Spring 2020. (**a**) Height of individuals of *A. berlengensis* growing in the exclusion areas A(1–3) “structure + *Armeria*”. (**b**) Height of individuals of *A. berlengensis* growing in the control areas A(4–6) “No structure + *Armeria*”. (**c**) Diameter of individuals of *A. berlengensis* growing in the exclusion areas A(1–3) “No structure + *Armeria*”. (**d**) Diameter of individuals of *A. berlengensis* growing control areas A(4–6) “No structure + *Armeria*”. ◦ represent outliers and * represent extreme outliers (that is, even more extreme outliers).

**Figure 7 plants-10-00498-f007:**
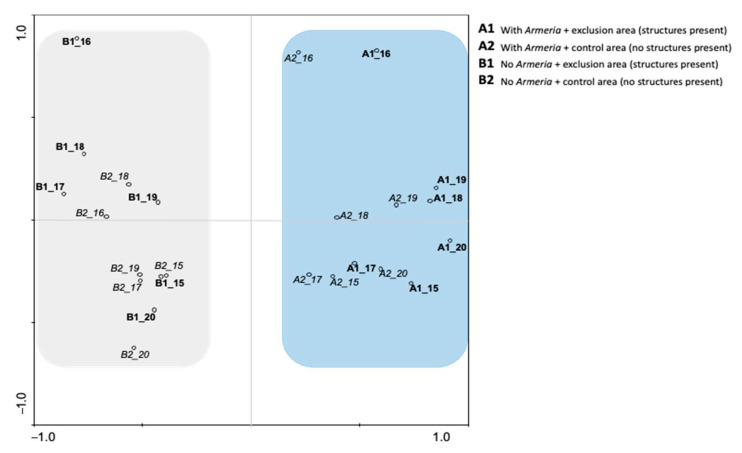
PCA biplot showing the PC1, with a clear dissimilarity between the areas presenting *A. berlengensis* (A1 and A2) and those without any *A. berlengensis*, (B1 and B2) for all the sampling period (Spring 2015–Spring 2020).

**Figure 8 plants-10-00498-f008:**
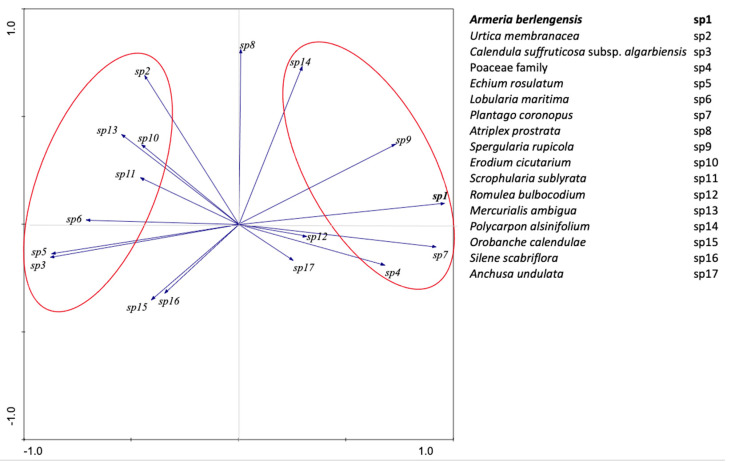
PCA biplot for the distribution of the most abundant species of Berlengas island, for all the sampling period (Spring 2015–Spring 2020).

**Figure 9 plants-10-00498-f009:**
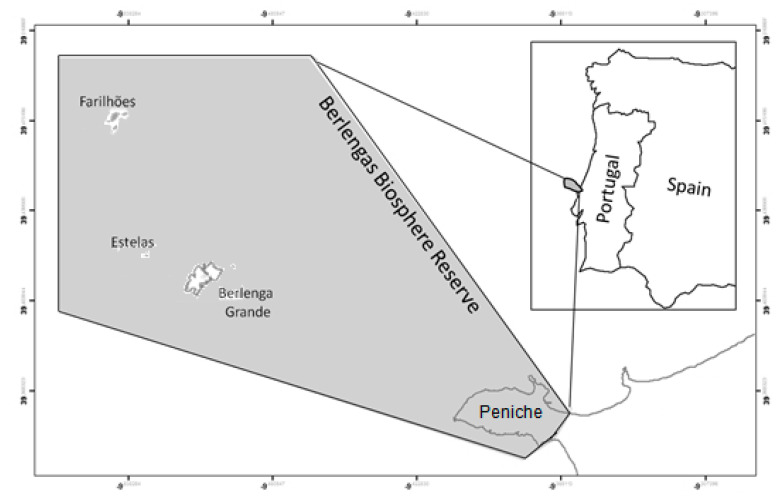
Berlengas archipelago location.

**Figure 10 plants-10-00498-f010:**
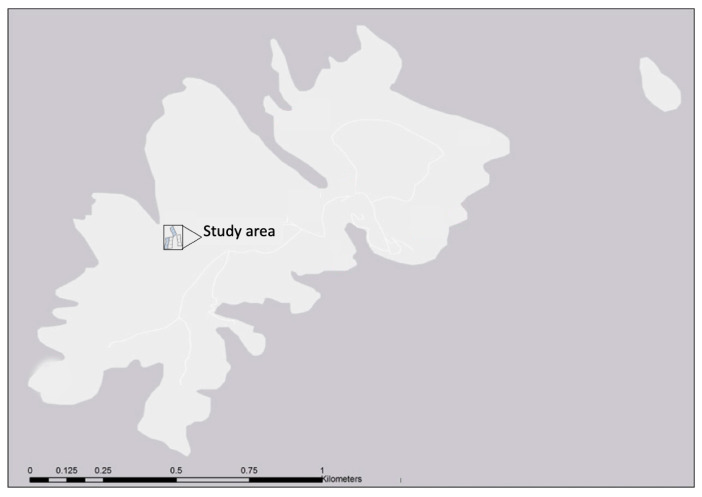
Location of the study area in Berlengas main island.

**Figure 11 plants-10-00498-f011:**
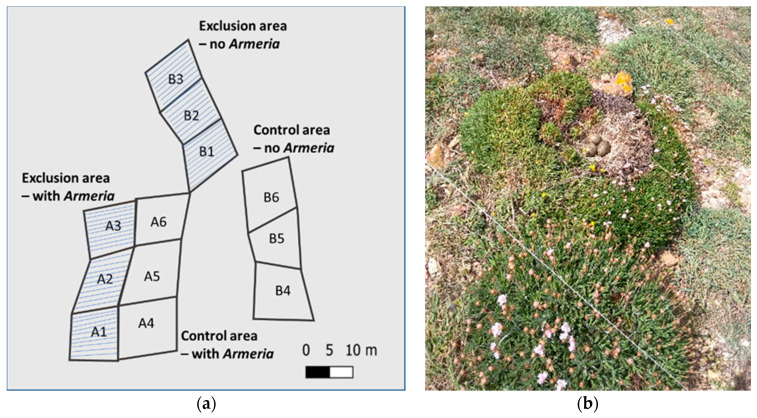
(**a**) Study areas defined in the plateau of Berlengas Island: “structure + *Armeria*” A(1–3): exclusion area with *A. berlengensis* growing; “no structure + *Armeria*” A(4–6): no structures installed with *A. berlengensis* growing; “structure + no *Armeria*” B(1–3) exclusion area without any *A. berlengensis*; “no structure + no *Armeria*” B(4–6): no structures installed in an area without any *A. berlengensis*. (**b**) Nest of a Yellow-legged Gull on an adult *A. berlengensis*, summer 2017, exclusion zone A3. The partial destruction of the plant is easily seen, both due to gull-tramping and to nitrophilous excrements.

**Figure 12 plants-10-00498-f012:**
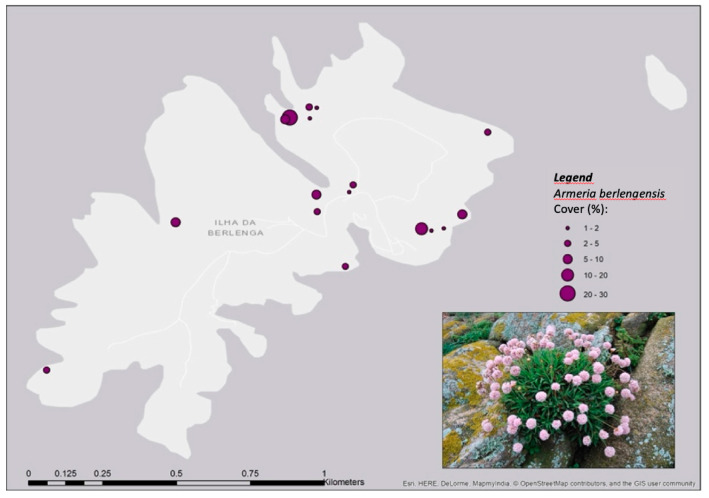
Distribution of *Armeria berlengensis* on Berlengas island, data collected from 2016–2018. Adapted from [16].

**Table 1 plants-10-00498-t001:** Total number of *A. berlengensis* individuals in the exclusion areas A(1–3) “Structure + *Armeria*”, and in the control areas A(4–6)”No structure + *Armeria*” and B(4–6) “No structure + no *Armeria*”, counted between the spring of 2015 (“spring15”) and the spring of 2020 (“spring20”).

Total Number of Individuals	Spring15	Spring16	Spring17	Spring18	Spring19	Spring20
A(1–3): Structure + *Armeria*	85	88	90	96	118	201
A(4–6): No structure + *Armeria*	7	6	8	5	6	17
B(4–6): No strucuture + no *Armeria*	1	0	0	0	0	0
Total	93	94	98	101	124	218

**Table 2 plants-10-00498-t002:** Summary of PCA of the exclusion areas and the control areas.

	1	2	3	4	Total Variance
Eigenvalues	0.367	0.149	0.119	0.083	1
Cumulative percentage variance of species data	36.7	51.6	63.5	71.8	--

**Table 3 plants-10-00498-t003:** Total number of *A. berlengensis* fruit clusters opened, and seeds recovered.

	Data Analyzed	#
Fruits	Total number of fruits opened	4721
	Mean number of fruits per cluster	48.17
Seeds	Total number of seeds collected	164
	Mean number of seeds collected	1.67
	Seed set	3.47%

**Table 4 plants-10-00498-t004:** Seed set of *Armeria berlengensis*, percentage of empty calyxes, non-fertilized ovaries, and rotten calyxes observed in 19 infructescences observed in detail.

	#	%
Number of fruit clusters analyzes	19	--
Seed set	23	3.39
Empty calyxes	207	30.53
Non-fertilized ovaries	343	50.59
Rotten calyxes	105	15.49
Total number of calyxes	678	--

## Data Availability

The research has been conducted in compliance with the Convention on Biological Diversity and the Convention on the Trade in Endangered Species of Wild Fauna and Flora. Specimens of *A. berlengensis* are deposited in the Herbarium of the University of Coimbra, Portugal (COI00084706, COI00084707, COI00084687).

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
