# Peer review of "Recent Efforts to Recover Armeria berlengensis, an Endemic Species from Berlengas Archipelago, Portugal"

_plants, 2021, doi:10.3390/plants10030498_

Round 1
Reviewer 1 Report
In the manuscript entitled "Recent efforts to recover Armenia berlengensis, an endemic species from Berlengas archipelago, Portugal", the authors presented the results research on protective activities of the local population of endemic plant species on the islands belonging to the Marine Reserve Area, covered by UNESCO World Heritage. Extensive conservation efforts have been carried out in this area for years, as Armeria berlengensis is now a critically endangered species. The efforts to prevent its extinction certainly require considerable effort and cost. A huge expense was probably incurred for the removal of Carpobrotus edulis, the invasive species of African origin. Of course, it was not the only protective action, I am listing it only as an example activity. In mat 2015 the study area was established in the plateau mentioned as the Valley of Armeria. The experimental setup is clear and correctly described and the data that could be subject to a statistically analysis was subjected to such analysis. Statistical analyzes were properly performed. In the course of the study, the vegetation in the selected plots was assessed. The presence of nitrolile species was correctly checked as the islands are a breeding ground for birds. A. berlengensis was subjected to a detailed and quite comprehensive analysis. I kindly ask the authors to refine the quality of figures (from 3d to 6), and in line 68 (m2). 2 should be increased to the power. The presented results are interesting and the work will certainly be continued over the years. When improving the work, please also think through the conclusions so that they correspond only with the results obtained. I think that they will gain clarity, which will facilitate their readability. However, I do not impose "only the right solution" here, I leave the decision to the authors on how to approach it. To sum up, I rarely have the pleasure of reviewing a text so valuable in terms of its importance for nature protection.
Author Response
The authors thank the reviewer for the valuable comments.
Improved figures were provided.
As to the conclusions, we did not make significantly changes, to try to accommodate the comments of the other reviewers. Only more detailed information was added regarding the results obtained.
Reviewer 2 Report
This manuscript should be re-written and improved to be considered for publication in Plants. My main comments are below.
The title looks to be well-written by reflecting the main contents.
The abstract doesn't reflect well the contents of the manuscript. Now, it includes the description of the study design and research backgrounds. However, there is a lack of the main results obtained. In addition, I recommend to add some conservation implications or recommendations into the abstract.
The list of key words must be revised considerably. At first, please, exclude the words and sentences presented in the title. At second, please, add more than three key words. I recommend to use at least five key words.
The section Introduction should be considerably improved,
First of all, authors should highlight the international relevance of research of this topic, but not be focused on the study area only. This makes this research to be of local level. I recommend to involve references published in the international journals.
But now a considerable part of the section Introduction is a description of study area. Authors must move these fragments into the section Material and Methods (subsection Study area).
In lines 88-90, please, note that if you mention IUCN Red List status, you should write "Vulnerable", but not "vulnerable", and in the same way for other categories.
Fig.2 should be moved into the section Material and Methods, too, as a description of sampling size.
Test of lines 98-110 is a description of the studied species, and it should be also moved into the section Material and Methods in subsection Study species.
At the last paragraph of the section Introduction, there is unclear information. Authors stated two aims (one is "goal", second is "aim"). Please, note that you should state one aim of the study, and several research tasks serving to deal the stated aim, as well as you may add hypotheses.
The section Material and Methods looks well, but it is incomplete. As it was stated above, I recommend to add some fragments into this section from Introduction.
The section Results is relatively well written. Most of this section is written in an appropriate way. However, PCA analysis and its description needs to be corrected. For example, in 278-280, authors state that Fig.10 indicate the most abundant species. But the presented plot doesn't reflect the abundance values. I suggest to use another way for this. For example, cluster analysis or other tests. Actually, this plot indicates the inter-associations of species in terms of abundance values. I think that after a such revision this result would be much better.
Concerning Armeria berlengensis seed set, I would suggest trying to test these results or trying to find correlation between seed number and measured morphometric traits, like height and diameter of the individuals. It would be better, I suppose.
The section Discussion should be revised, too. So, the first paragraph is rather description of results presented in the previous section. And it is rather a repetition of the previously mentioned information. I suggest to omit it here.
Next, I don't recommend to use the such title as Vegetation survey because authors didn't study vegetation, i.e. plant communities. Authors studied plants growing together with the target species. Also, I don't recommend to name the next subsection (PCA analysis) as a separate subsection. I recommend to not distinguish the Discussion by subsections.
In general, I advise authors to compare the obtained results with other references published previously. Of course, I don't mean that these references should be devoted to the same plant species, but I recommend to involve the published data (of international level) of plants having the similar biology and ecology.
In my view, the section Conclusions seems to be of good quality and contents. Only one thing, which might be added, is addition of more detailed information.
Author Response
Comments to review n. 2.
The authors thank the valuable comments and suggestions made by the reviewer, that have much improved the manuscript.
The manuscript has been fully revised, as suggested.
All the changes were made in the manuscript as track changes, in the attached document.
Regarding the abstract:
“The abstract doesn't reflect well the contents of the manuscript.Now, it includes the description of the study design and researchbackgrounds. However, there is a lack of the main resultsobtained. In addition, I recommend to add some conservationimplications or recommendations into the abstract. The list of key words must be revised considerably. At first, please, exclude the words and sentences presented in the title.At second, please, add more than three key words. I recommendto use at least five key words.”
ANSWER: The abstract can only hold 200 words, making it impossible to include all the information requested, but we revised the abstract and included the main results obtained.
Introduction:
“The section Introduction should be considerably improved, First of all, authors should highlight the international relevance ofresearch of this topic, but not be focused on the study area only.This makes this research to be of local level. I recommend toinvolve references published in the international journals.”
ANSWER: Changes were made in the introduction to include general information on oceanic islands and their endemic species, as well as the main threats these plants undergo. New international references were added, as requested.
“But now a considerable part of the section Introduction is adescription of study area. Authors must move these fragmentsinto the section Material and Methods (subsection Study area).”
ANSWER: Sections of the introduction were moved to Materials and Methods.
The aim and tasks of the work has been rewritten, as requested.
“In lines 88-90, please, note that if you mention IUCN Red List status, you should write "Vulnerable", but not "vulnerable", and inthe same way for other categories“
ANSWER: words have been changed to substitute to capital letters.
“Fig.2 should be moved into the section Material and Methods,too, as a description of sampling size. Test of lines 98-110 is a description of the studied species, and itshould be also moved into the section Material and Methods insubsection Study species.”
ANSWER: Section has been moved to Materials and Methods.
“At the last paragraph of the section Introduction, there is unclearinformation. Authors stated two aims (one is "goal", second is"aim"). Please, note that you should state one aim of the study,and several research tasks serving to deal the stated aim, aswell as you may add hypotheses.”
ANSWER: Section of the introduction was revised to include one aim and several tasks.
As to Material and Methods:
The section Material and Methods looks well, but it isincomplete. As it was stated above, I recommend to add somefragments into this section from Introduction.
ANSWER: the suggested sentences of the introduction were added to this section.
As to results:
"However, PCA analysis and its description needs to be corrected. For example, in 278-280, authors state that Fig.10 indicate the most abundant species. But the presented plot doesn't reflect the abundance values. I suggest to use another way for this. For example, cluster analysis or other tests. Actually, this plot indicates the inter-associations of species in terms of abundance values."
ANSWER: The authors understand the observation. We used percentage of cover, as stated in the methods and thus we have rewritten the sentence as follows:
“Regarding the species coverage (%) pattern, the most characteristic rupicolous (...)”
Adttionally, the authors stress out that in the material and methods (statistical analysis section), the data matrix on which the PCA was performed is, we believe, correctly described and identified. In addition, PCA is a method of data reduction. It aims to reduce a large number of variables to a (much) smaller number while losing as little information as possible (Which meets what was intended with this analysis). The use of an analysis of clusters would be reductive, with regard to the pattern that is intended to be evidenced with the PCA. On the other hand, according to the other reviewers, the present analysis is correct, well-suited and responds to the objectives of the study.
“Concerning Armeria berlengensis seed set, I would suggest trying to test these results or trying to find correlation between seed number and measured morphometric traits, like height and diameter of the individuals. It would be better, I suppose.”
ANSWER: The authors cannot answer to your suggestion about seed set, for we do not have height nor diameter data on the plants the seeds were collected from. We only have the information that they were adults, higher than 30 cm. But, because we want to do germination tests, in the future, these data will be retrieved and analysed. Thank you so much for the suggestion.
As to Discussion:
“The section Discussion should be revised, too. So, the firstparagraph is rather description of results presented in theprevious section. And it is rather a repetition of the previouslymentioned information. I suggest to omit it here.”
ANSWER: It has been reviewed, including information on oceanic islands worldwide, and new international references were added, as requested.
“Next, I don't recommend to use the such title as Vegetationsurvey because authors didn't study vegetation, i.e. plantcommunities. Authors studied plants growing together with thetarget species. Also, I don't recommend to name the nextsubsection (PCA analysis) as a separate subsection. Irecommend to not distinguish the Discussion by subsections.”
ANSWER: Subsections have been removed.
In general, I advise authors to compare the obtained results with her references published previously. Of course, I don't meanthat these references should be devoted to the same plantspecies, but I recommend to involve the published data (ofinternational level) of plants having the similar biology andecology.
ANSWER: new international references were added, as requested, to compare the data from our case with other cases in islands, at international level, namely the following:
Vitousek, P.M. Diversity and biological invasions of oceanic islands. In Biodiversity; Wilson, E.O., Peter, F.M., Eds.; National Academies Press (US), 1988 ISBN 0-309-03783-2I.
Le Corre, M.; Danckwerts, D.K.; Ringler, D.; Bastien, M.; Orlowski, S.; Morey Rubio, C.; Pinaud, D.; Micol, T. Seabird recovery and vegetation dynamics after Norway rat eradication at Tromelin Island, western Indian Ocean. Biol. Conserv. 2015, 185, 85–94.
Scott, J.J.; Kirkpatrick, J.B. Rabbits, landslips and vegetation change on the coastal slopes of subantarctic Macquarie Island, 1980-2007: Implications for management. Polar Biol. 2008, 31, 409–419.
Canale, D.E.; Dio, V. Di; Massa, B.; Mori, E. First successful eradication of invasive Norway rats Rattus norvegicus from a small Mediterranean island (Isola delle Femmine, Italy). Folia Zool. 2019, 68, 29.
Eijzenga, H. Vegetation change following rabbit eradication on Lehua Island, Hawaiian Islands. In Proceedings of the Proceedings of the International Conference on Island Invasives; CR Veitch, C.R., Clout, M.N., Towns, D.R., Eds.; 2011; pp. 290–294.
Bell, B.D. The eradication of alien mammals from five offshore islands, Mauritius, Indian Ocean. In Turning the Tide: The Eradication of Invasive Species; Veitch, C.R., Clout. M.N., Eds.; IUCN: Gland, Switzerland and Cambridge, U.K., 2002; p. 414.
Rita, J.; Biblioni, G. The flora of the islets of the Balearic Islands. In Proceedings of the Islands and Plants: Preservation and Understanding of Flora on Mediterranean Islands,. 2nd Botanical Conference in Menorca. Proceedings and abstracts.; Cardona-Pons, E., Estaún-Clarisó, I., Comas-Casademont, M., I., F., Arguimbau, P., Eds.; Colleció recerca, 2013.
Tomassen, H.B.M.; Smolders, A.J.P.; Lamers, L.P.M.; Roelofs, J.G.M. How bird droppings can affect the vegetation composition of ombrotrophic bogs. Can. J. Bot. 2005, 83, 1046–1056.
Sigurdsson, B.D. Effects of sea birds and soil development on plant and soil nutritional parameters after 50 years of succession on Surtsey. Surtsey Res. 2020, 14, 85–90.
De La Peña-Lastra, S.; Gómez-Rodríguez, C.; Pérez-Alberti, A.; Torre, F.; Otero, X.L. Effects of a yellow legged gull (Larus michahellis) colony on soils and cliff vegetation in the Atlantic Islands of Galicia National Park (NW Spain). Catena 2021, 199, 105115.
As to the conclusions
“Only one thing, which might be added, is additionof more detailed information”
ANSWER: more detailed information was added about the results obtained.

Reviewer 3 Report
It is an interesting manuscipt, dealing with a rare endemic threatened species.
Important; The "Material & Methods" section must be moved before the "Results"
I suggest acceptance after minor revisions:
Page 1 line 31: Delete “Endemic species” (do not use in the keywords words that are already included in the Title)
Page 2 lines 63-64: “The endemic and halophytic flora of Berlengas’ Archipelago is threatened by several factors.” Here it would be useful to indicate some references to other studies dealing with the conservation status of endemics, also in other European territories. For example:
The endemic and halophytic flora of Berlengas’ Archipelago is threatened by several factors, as observed in many other European coastal territories which host threatened endemics with a very limited distribution area (e.g. Trias-Blasi et al. 2011; Foggi et al. 2015; Wagensommer et al. 2020).
Trias-Blasi, A.; Eddie, W.M.M.; Hedge, I.C.; Möller, M.; Sales, F. The taxonomy and conservation of Campanula primulifolia (Campanulaceae), a critically endangered species in the Iberian Peninsula. Willdenowia 2011, 41, 35-42. doi:10.3372/wi.41.41103.
Foggi, B.; Viciani, D.; Baldini, R.M.; Carta, A.; Guidi, T. Conservation assessment of the endemic plants of the Tuscan Archipelago, Italy. Oryx 2015, 49(1): 118-126. doi:10.1017/S0030605313000288.
Wagensommer, R.P.; Medagli, P.; Turco, A.; Perrino, E.V. IUCN Red List evaluation of the Orchidaceae endemic to Apulia (Italy) and conservations on the application of the IUCN protocol to rare species. Nature Conservation Research 2020, 5(suppl. 1), 90-101. doi:10.24189/ncr.2020.033.
Page 2 line 68: Instead of “m2” write “m2”
Page 3 lines 69-72: What species order did you follow? Importance? If yes, specify it. If not, follow alphabetic order.
Page 3, Figure 2: Instead of “Legenda” write “Legend” and instead of “Cobertura” write “Cover”
Page 4 line 135: This is not vegetation survey! Maybe you mean a survey of the Armeria population, but not vegetation.
Page 4 line 151 (from 85 to 215): In Table 1 the number ranges from 85 to 201. Check what is correct!
Page 7 line 200: Insert “individuals” between “all the” and “measured”: for all the individuals measured
Page 8, Figure 7: Specify the meaning of all the 3 different symbols used above the bars
Page 8, line 240: Instead of “isn’t” write “is not”
Page 9, line 251: “Boxplots” instead of “Boxplot”
Page 9, line 256: Explain the meaning of the 2 different symbols used
Page 10, line 282: What do you mean with “Opposing”? Explain better.
Page 10, 286-287: “Lobularia maritima (L.) Desv. subsp. maritima” instead of “Lobularia maritima subsp. maritima (L.) Desv.”
Page 10, line 288: This is the first mention of Mercurialis ambigua; therefore, indicate the author of the species
Page 10, line 289: This is the first mention of Erodium cicutarium; therefore, indicate the author of the species
Page 10, line 289: delete “and” before Scrophularia sublyrata Brot.
Page 11, line 311: “by observing at 19 infructescences”, maybe better “by observation on 19 infructescences”
Page 13, line 400: What do you mean with “is opposed”. Explain better
Page 13, line 408: This is the first mention of O. calendulae; therefore, indicate the author of the species and write the whole genus name
Page 14, line 426: “4.3. Seed set” in italics
Page 14: Move the “Materials and Methods” section before the “Results” section (page 4), changing the numbering: 2. Materials and Methods; 2.1. Surveys; 2.2 Statistical analysis; 3. Results; 3.1. Vegetation surveys
Page 14, line 450: instead of “is declining” write “was declining”
Page 15, lines 476-477: What do you mean with “vegetation assessment”? Phytosociological relévés? In the Results and Discussion you do not report and discuss the results of these vegetation assessments.
Page 18 line 596: Do not use initial capital letter for specific epithets: Carpobrotus edulis; Malcolmia littorea (In addition: Why binomial names not in italics?)
Page 18 line 601: Do not use initial capital letter for specific epithets: Armeria berlengensis (In addition: Why binomial name not in italics?)
Throughout the text: You used 15-times the word “specimens” and 1-time the word “specimen”. I suggest to write “individuals/individual” (or “ramets/ramet”?) instead of “specimens/specimen”
Author Response
The authors thank the valuable comments and suggestions made by the reviewer, that have much improved de manuscript.
The manuscript has been fully revised, namely the introduction, as suggested.
Regarding your comment: Important; The "Material & Methods" section must be moved before the "Results"
Important; The "Material & Methods" section must be moved before the "Results"
ANSWER: The authors understand the remark, but the Plants template places the “4. Materials and Methods” after the “3. Discussion” and there is no flexibility regarding the structure of the manuscript.
Page 1 line 31: Delete “Endemic species” (do not use in the keywords words that are already included in the Title)
ANSWER: The authors agree with the reviewer and have changed the keywords.
Page 2 lines 63-64: “The endemic and halophytic flora of Berlengas’ Archipelago is threatened by several factors.” Here it would be useful to indicate some references to other studies dealing with the conservation status of endemics, also in other European territories. For example:
The endemic and halophytic flora of Berlengas’ Archipelago is threatened by several factors, as observed in many other European coastal territories which host threatened endemics with a very limited distribution area (e.g. Trias-Blasi et al. 2011; Foggi et al. 2015; Wagensommer et al. 2020).
Trias-Blasi, A.; Eddie, W.M.M.; Hedge, I.C.; Möller, M.; Sales, F. The taxonomy and conservation of Campanula primulifolia (Campanulaceae), a critically endangered species in the Iberian Peninsula. Willdenowia 2011, 41, 35-42. doi:10.3372/wi.41.41103.
Foggi, B.; Viciani, D.; Baldini, R.M.; Carta, A.; Guidi, T. Conservation assessment of the endemic plants of the Tuscan Archipelago, Italy. Oryx 2015, 49(1): 118-126. doi:10.1017/S0030605313000288.
Wagensommer, R.P.; Medagli, P.; Turco, A.; Perrino, E.V. IUCN Red List evaluation of the Orchidaceae endemic to Apulia (Italy) and conservations on the application of the IUCN protocol to rare species. Nature Conservation Research 2020, 5(suppl. 1), 90-101. doi:10.24189/ncr.2020.033.
ANSWER: The authors agree with the reviewer and have changed the introduction including the suggested references and also included:
Baumel, A.; Youssef, S.; Ongamo, G.; Medail, F. Habitat suitability assessment of the rare perennial plant Armeria arenaria (Pers.) Schult. (Plumbaginaceae) along the French Mediterranean coastline. Candollea 2013, 68, 221–228.
The introduction was revised to include a more international view of the problem.
Page 2 line 68: Instead of “m2” write “m2”
ANSWER: Thank you, the mistake has been corrected in the attached manuscript.
Page 3 lines 69-72: What species order did you follow? Importance? If yes, specify it. If not, follow alphabetic order.
ANSWER: Thank you, alphalbetic orther has been followed..
Page 3, Figure 2: Instead of “Legenda” write “Legend” and instead of “Cobertura” write “Cover”
ANSWER: Thank you, the mistake has been corrected in the attached manuscript.
Page 4 line 135: This is not vegetation survey! Maybe you mean a survey of the Armeria population, but not vegetation.
ANSWER: Thank you, it mistake has been corrected in the attached manuscript to “Armeria berlengensis surveys”
Page 4 line 151 (from 85 to 215): In Table 1 the number ranges from 85 to 201. Check what is correct!
ANSWER: Thank you, the mistake has been corrected in the attached manuscript (201 is the correct number).
Page 7 line 200: Insert “individuals” between “all the” and “measured”: for all the individuals measured
ANSWER: Thank you, the mistake has been corrected in the attached manuscript.
Page 8, Figure 7: Specify the meaning of all the 3 different symbols used above the bars
ANSWER: Thank you, symbols have been specified in the caption of the figure.
Page 8, line 240: Instead of “isn’t” write “is not”
ANSWER: Thank you, the mistake has been corrected in the attached manuscript.
Page 9, line 251: “Boxplots” instead of “Boxplot”
ANSWER: Thank you, the mistake has been corrected in the attached manuscript.
Page 9, line 256: Explain the meaning of the 2 different symbols used
ANSWER: Thank you, symbols have been specified in the caption of the figure.
Page 10, line 282: What do you mean with “Opposing”? Explain better.
ANSWER: the sentence has been rephrased to “In the CP1 negative axis there is another cluster which includes the ruderal nitrophilous species such as C. suffruticosa subsp. algarbiensis (sp3) and Echium rosulatum (sp5).” We hope it is more understandable now.
Page 10, 286-287: “Lobularia maritima (L.) Desv. subsp. maritima” instead of “Lobularia maritima subsp. maritima (L.) Desv.”
ANSWER: Thank you, the mistake has been corrected in the attached manuscript.
Page 10, line 288: This is the first mention of Mercurialis ambigua; therefore, indicate the author of the species
ANSWER: Thank you, the mistake has been corrected in the attached manuscript.
Page 10, line 289: This is the first mention of Erodium cicutarium; therefore, indicate the author of the species
ANSWER: Thank you, the mistake has been corrected in the attached manuscript.
Page 10, line 289: delete “and” before Scrophularia sublyrata Brot.
ANSWER: Thank you, the mistake has been corrected in the attached manuscript.
Page 11, line 311: “by observing at 19 infructescences”, maybe better “by observation on 19 infructescences”
ANSWER: Thank you, the mistake has been corrected in the attached manuscript.
Page 13, line 400: What do you mean with “is opposed”. Explain better
ANSWER: the sentence has been rephrased to “PCA results also confirm that A. berlengensis and the other rupicolous species are on the CP1 positive axis while C. suffruticosa subsp. algarbiensis and E. rosulatum are on the negative PC1 axis.”. We hope it is more understandable.
Page 13, line 408: This is the first mention of O. calendulae; therefore, indicate the author of the species and write the whole genus name
ANSWER: Thank you, the mistake has been corrected in the attached manuscript.
Page 14, line 426: “4.3. Seed set” in italics
ANSWER: subsections have been deleted, as advised by reviewer n.2.
Page 14: Move the “Materials and Methods” section before the “Results” section (page 4), changing the numbering: 2. Materials and Methods; 2.1. Surveys; 2.2 Statistical analysis; 3. Results; 3.1. Vegetation surveys
ANSWER: As stated, the Plants template places the “4. Materials and Methods” after the “3. Discussion”.
Page 14, line 450: instead of “is declining” write “was declining”
ANSWER: Thank you, the text has been changes to “species presence and cover were assessed”.
Page 15, lines 476-477: What do you mean with “vegetation assessment”? Phytosociological relévés? In the Results and Discussion you do not report and discuss the results of these vegetation assessments.
ANSWER: Thank you, the mistake has been corrected in the attached manuscript.
Page 18 line 596: Do not use initial capital letter for specific epithets: Carpobrotus edulis; Malcolmia littorea (In addition: Why binomial names not in italics?)
Page 18 line 601: Do not use initial capital letter for specific epithets: Armeria berlengensis (In addition: Why binomial name not in italics?)
ANSWER: Thank you, references were fully reviewed to correct these mistakes.
Throughout the text: You used 15-times the word “specimens” and 1-time the word “specimen”. I suggest to write “individuals/individual” (or “ramets/ramet”?) instead of “specimens/specimen”
ANSWER: Thank you, specimens were corrected to individuals, as suggested.

Round 2
Reviewer 2 Report
Unfortunately, authors didn't make appropriate revisions, although they stated that did it. For example, the description of the study area is still presented in the Introduction section.
Actually, authors have moved only the description of the studied species into the section Material and Methods. Moreover, now the description of Armeria berlengensis is mentioned under the sub-section Study area. Obviously, it is inappropriate.
Concerning Abstract, although authors stated that they re-wrote this section, again I see that there is no aim of the study, main data on material and methods, while there are irrelevant fragments, which could be omitted. Therefore, the abstract still needs in revision.
Although authors stated that they added research tasks, actually, they stated what did they study during the survey. Actually, this information is rather a thesis of results obtained, but not tasks established before at the stage of the survey planning.
I think that if authors cannot test the results, they should not say about reliable results, because the test have not been conducted. In this case, I recommend to remove these sub-sections of the section Results, with testing these data in future (in the further publications). But, as authors so insist on that, I think that it is up to choice of Editors.
Thus, the revision has rather cosmetic character, without paying an attention to many comments.
Author Response
The authors thank the comments and suggestions made by the reviewer, which we have tried to integrate into the text of the article.
Unfortunately, authors didn't make appropriate revisions, although they stated that did it. For example, the description of the study area is still presented in the Introduction section.
ANSWER: The description of the study area has been moved to the Material and Methods, as requested.
Actually, authors have moved only the description of the studied species into the section Material and Methods. Moreover, now the description of Armeria berlengensis is mentioned under the sub-section Study area. Obviously, it is inappropriate.
ANSWER: A new subsection has been added for the description and distribution of the species: 4.2. Armeria berlengensis morphology and distribution
Concerning Abstract, although authors stated that they re-wrote this section, again I see that there is no aim of the study, main data on material and methods, while there are irrelevant fragments, which could be omitted. Therefore, the abstract still needs in revision.
ANSWER: The abstract has been reviewed, including, as requested, more information on the aim of the study, main conclusions, and conservation measures.
"Berlengas’ archipelago is an UNESCO’s world heritage site, the only location where Armeria berlengensis is found. This species faces some threats, namely human disturbance, the presence of Carpobrotus edulis, and the Yellow-legged Gull, Common-rabbit, and Black-rat populations. Thus, exclusion areas were installed which blocked the access of most Gulls, aiming to promote the recovery of A. berlengensis. Besides, rabbits and rats were removed from the island. After six years of surveys, there is an increase in the number of individuals of A. berlengensis in the exclusion areas, and a clear shift in the size structure of the A. berlengensis population. Significant changes in the height and diameter of the individuals were also noted. These findings indicate that the population of A. berlengensis is changing to a healthier population. PCA results show a straightforward dissimilarity between the areas with A. berlengensis and those without the species and allowed the clustering of two groups: the rupicolous species and the nitrophilous species. A. berlengensis produces few seeds (seed set 3,4%), which raises concern regarding the long-term survival of the species. Thus, further conservation efforts must be implemented, such as control of invasive species, Gulls, and ruderals, to allow the recovery of A. berlengensis."
Although authors stated that they added research tasks, actually, they stated what did they study during the survey. Actually, this information is rather a thesis of results obtained, but not tasks established before at the stage of the survey planning.
ANSWER: The authors are not sure if they understood this remark, but the text has been rewritten: “The height and diameter of A. berlengensis was also determined annually to evaluate the growth of the individuals and the size structure of the A. berlengensis population in the studied areas. Moreover, we evaluated the number of seeds produced by A. berlengensis, by seed cluster, to estimate the seed set of the species, aiming to understand the recovery capacity of the species.”
I think that if authors cannot test the results, they should not say about reliable results, because the test have not been conducted. In this case, I recommend to remove these sub-sections of the section Results, with testing these data in future (in the further publications). But, as authors so insist on that, I think that it is up to choice of Editors.
ANSWER: We believe that this remarks of the reviewer refers to seed set. As stated in the previous review, we believe that these data are relevant for the paper, because the low seed set found may hinder the recovery of A. berlengensis. Yet, unfortunately, we could not add the new data requested by the reviewer because, as stated, “we do not have height nor diameter data on the plants the seeds were collected from. We only have the information that they were adults, higher than 30 cm. But, because we want to do germination tests, in the future, these data will be retrieved and analyzed.”
